# The Impact of SARS-CoV-2 Immune Status and Societal Restrictions in Controlling COVID-19 across the World

**DOI:** 10.3390/vaccines11091407

**Published:** 2023-08-23

**Authors:** Jasmijn Stroo, Michaëla Lepolder, Jean-Luc Murk, Ger T. Rijkers

**Affiliations:** 1Science Department, University College Roosevelt, 4331 CB Middelburg, The Netherlands; j.stroo@student.maastrichtuniversity.nl (J.S.); m.lepolder@student.maastrichtuniversity.nl (M.L.); 2Microvida Laboratory for Medical Microbiology and Immunology, St. Elisabeth Hospital, 5022 GC Tilburg, The Netherlands

**Keywords:** COVID pass, SARS-CoV-2 antigen test, SARS-CoV-2 antibodies, societal restrictions

## Abstract

To control the COVID-19 pandemic, many countries implemented vaccination and imposed societal restrictions both at the national level and for international travel. As a check of corona status, COVID passes have been issued. A COVID pass could be obtained when either fully vaccinated against COVID-19, or after recovering from a documented COVID-19 episode, or after a recent (24–48 h) negative SARS-CoV-2 antigen test. A global analysis of SARS-CoV-2 immune status determined by past infection and/or vaccination, vaccination rates, as well as societal restrictions in controlling the COVID-19 pandemic is presented. The data show that across the world, vaccination was more effective in reducing SARS-CoV-2 infections with the delta variant than the omicron variant. Strict societal restrictions could control spread of the virus, but relief of the restrictions was associated with an increase in omicron infections. No significant difference in SARS-CoV-2 infections were found when comparing countries or territories which did or did not implement a COVID pass.

## 1. Introduction

Coronavirus disease 2019 (COVID-19) is an infectious disease caused by severe acute respiratory syndrome coronavirus 2 (SARS-CoV-2) [1]. On 30 January 2020 the World Health Organization (WHO) officially declared the SARS-CoV-2 outbreak a Public Health Emergency of International Concern. About a month and a half later, on 11 March 2020, this outbreak was formally declared a global pandemic [2,3]. The WHO urged countries to adopt strict social distancing and other (quarantine) measures to protect public health by preventing virus spread [4,5]. As of June 2022, SARS-CoV-2 has over 500 million confirmed infections, including over six million deaths reported to WHO [6].

Handling the global COVID-19 crisis has been a multifactorial operation requiring the coordinated action of all levels of healthcare, government, pharmaceutical industries, and non-government organizations, preferably with international consultation and coordination [7]. Although the mechanisms of viral transmission, infection, and treatment were largely unknown during the early phases of the pandemic, action from physicians, scientists, and governments was urgently needed to prevent further spread. While every country had its own unique approach to limit further infections, common threads in the initial response were interventions such as school and workplace closures, cancellation of public events and gatherings, stay-at-home restrictions, face coverings, and (international and domestic) travel restrictions [7,8]. With time, the understanding of the virus grew, and with that also the development of (public) testing facilities and treatment. Within a year after the outbreak of the pandemic and the identification of the genomic structure of SARS-CoV-2, several highly effective vaccines were approved and used globally, as nearly 12 billion vaccine doses have been administered (dated June 2022; World Health Organization) [6,9,10,11].

The availability of an effective vaccine and fluctuation in the number of infections made governments allow for changes in societal restrictions, increasing or decreasing the intensity of these restrictions depending on what was needed to control the virus. Many countries have introduced a ‘corona pass’, or COVID pass [12]. The precise conditions under which this pass could be obtained varied among countries, but mostly included being fully vaccinated against COVID-19, having recovered from a documented COVID-19 episode, or a recent (24–72 h) negative SARS-CoV-2 antigen or PCR test [13]. Such a pass has allowed individuals to travel internationally, but often was also used on a national level, i.e., for access to indoor spaces such as bars and restaurants, theaters and museums, or other (large-scale) events [12,13,14].

As mentioned before, the world has displayed a diverse and fragmented approach to the preservation of public health and prevention of spread of COVID-19. In March 2021, Hale and colleagues introduced the Oxford COVID-19 Government Response Tracker (OxCGRT), a continuously updated, readily usable database on global policy measures [15]. Starting 1 January 2020, the data capture government policies related to closure and containment, health, and economic policies for more than 180 countries or territories. Policy responses are recorded ordinally or continuously for 19 policy areas, capturing variation in the degree of response [15]. Ultimately, the sum of the policy areas is calculated, resulting in an overall Government Stringency Index (GSI). Important is to note that economic measures, such as governmental support, are not used in the index calculations [15]. The OxCGRT enabled us to explore the empirical effect of policy responses on the spread of COVID-19 cases and vaccination status from a global viewpoint, with emphasis on the comparison between high, medium, and low-income countries across the world.

## 2. Methods

This paper aims to provide a global analysis of societal restrictions, vaccination status, and use of the COVID pass, thereby covering a wide spectrum of income status between countries which were included. Data was collected from the OxCGRT database and analyzed per continent, of which the selection of individual countries and territories was determined based on the coverage of the HDI and availability of sufficient and reliable data. In this research the following 42 countries and territories were analyzed: Asia: Azerbaijan; Bahrain; China; Cyprus; Georgia; India; Israel; Japan; Malaysia; Pakistan; Qatar; Saudi Arabia; South Korea; the Chinese territory Taiwan; Thailand; Türkiye; United Arab Emirates; and Vietnam. Africa: Egypt; Morocco; Nigeria; South Africa; Tunisia; Uganda; and Zambia. Europe: Belgium; Denmark; France; Hungary; Italy; Netherlands; Poland; Portugal; Spain; Sweden; and United Kingdom. America: Argentina; Brazil; Canada; Colombia; Mexico; and the United States.

For each country or territory the following parameters were obtained from https://ourworldindata.org/coronavirus (accessed on 31 March 2022): the peak infections per million people during the first wave, delta variant wave, and omicron B.1.1.529 variant wave; the maximum GSI of the entire pandemic and the GSI per March 2022; and vaccination rate (fully vaccinated) per 31 March 2022 [7]. It is important to note that there were differences in the time period during which the peak of each variant’s wave was reached. For the first wave dates ranged from March 2020 to September 2020, for the delta wave and omicron wave May 2021 to December 2021 and December 2021 to March 2022, respectively.

### 2.1. Government Stringency Index

The Government Stringency Index (GSI) is composed of nine individual components and includes school and workplace closures, cancellations of public events, restrictions on gathering sizes, closures of public transport, stay-at-home requirements, restrictions on internal movement and international travel, and public information campaigns. The individual components are aggregated into a single index which consists of a number ranging from 0–100 [15].

To determine whether a country or territory made use of a COVID pass and under what circumstances, functional publicly-available data from the official national health institutes was used. If such information was not stated by the national institute, secondary sources were used.

The Human Development Index (HDI) was retrieved from Human Development Reports, namely https://hdr.undp.org/data-center/human-development-index (accessed on 31 March 2022) and was based on the situation in 2020 [16]. The GDP per capita in the US $ was retrieved from the data of The World Bank, namely https://data.worldbank.org/ (accessed on 31 March 2022) and was based on 2020 [17]. Of this same source, the overall population density of a country (people per square kilometer of land area) was also determined [17].

### 2.2. Statistical Analysis

The significance of differences in Government Stringency Index during the Wuhan and the omicron wave, as well as the impact of introducing a COVID pass on the number of COVID-19 cases, was calculated by using two-sided Student’s *t*-test. Linear regression analysis was performed with GraphPad for https://www.graphpad.com/quickcalcs/linear1/ (accessed on 31 March 2022).

## 3. Results

In our study we have selected countries from across the globe for which sufficient, reliable data were available. Figure 1 shows the geographical location of these countries, color-coded by continent. For each country, the cumulative registered number of COVID-19 cases and deaths since the beginning of the pandemic is indicated (see also Table 1). The highest mortality, relative to the number of cases, was observed in Georgia (Asia), Mexico (America), and Hungary (Europe). Asia and Africa show the greatest variation between countries in the number of COVID-19 cases and deaths. Whether these differences are due to under-registration or under-reporting cannot be concluded from the available sources.

The total population of the countries and territories included in our study is 5.2 × 10^9^, which amounts to 65% of the world population. Although virtually all countries have reported the number of COVID-19 confirmed cases and deaths, these may not truly reflect the actual numbers. This can be due to limited testing capacity, which results in a lower number of confirmed cases than the true number of infections. Additionally, the number of confirmed deaths may be an underestimate of the true number of deaths caused by COVID-19 because of the use of varying protocols and the attribution of COVID-19 as the cause of death.

We find a significant positive correlation between the number of COVID-19 cases and the GDP per capita. The two outliers are China and Taiwan (indicated with blue symbols in Figure 2), which both show an (extremely) low number of COVID-19 cases while having a relatively high GDP per capita.

### 3.1. Prevention of SARS-CoV-2 Infection by Vaccination

For each continent, a positive correlation was found between the adult vaccination rate and the number of reported infections during the period when the delta variant of SARS-CoV-2 was dominant (Figure 3). The exceptions are African countries, which have low vaccination rates but also report relatively few infections.

Our data on the effect of vaccination were analyzed by comparing overall vaccination rates per country and continent with the overall number of reported SARS-CoV-2 infections. It should be stressed that the impact of vaccination within a given country would more clearly show the protective effect of vaccination (in terms of hospitalization, intensive care admission, and deaths), but this would preclude comparison between countries and continents.

### 3.2. Societal Restrictions

During the pandemic, virtually all countries have imposed, with varying degrees, societal restrictions. These restrictions were highest during the first period of the pandemic and gradually were lifted when vaccination had been implemented and when the omicron strain became prevalent. We quantified the partial lifting of the restrictions by comparing the difference between the situation during the first wave (Wuhan) and omicron. The data shown in Figure 4 indicates that the European countries where most of the restrictions had ended by the time omicron became dominant, showed the highest number of infections. In Asia and America, where restrictions were still higher when omicron emerged, a lower number of infections occurred. It should be noted that the data in Figure 4 show omicron infections, as measured by PCR or an antigen (self) test, and not COVID-19 hospital admissions.

### 3.3. Introduction and Effect of a COVID Pass

In most countries a COVID pass was introduced in 2021, ranging from January 2021 (Taiwan) to December 2021 (Tunisia, Japan and Georgia). For most countries or territories, a COVID pass was issued when a person recovered from a documented COVID-19 episode or was fully vaccinated (depending on the particular SARS-CoV-2 vaccine used this meant one or two doses) or had tested negative in a SARS-CoV-2 PCR test or antigen test. A negative test result was valid for 24–72 h. A COVID pass could give the carrier access to public places like bars, restaurants, and (movie) theaters and/or certain travel privileges such as travel by public transportation and across borders.

Apart from Asian countries and territories during the delta wave, no significant differences were found in the number of cases between countries that had introduced a COVID pass, and those that had not (Figure 5). A meaningful interpretation of these data however is hampered because data on the number of hospitalizations, as well as COVID deaths could not be compared. Furthermore, in many cases it was not clear when a COVID pass was introduced, under what conditions a COVID pass could be issued, or for which social or other public activities the COVID pass was required.

## 4. Discussion

The aim of this paper was to analyze the role of SARS-CoV-2’s serological status in societal restrictions across the world. As a proxy for serological status, we used the vaccination rate in a given country. To do so, the effect of GDP, HDI and GSI differences on the impact of the different SARS-Cov-2 variants was approximated for several countries representative of their continent. The continent of Oceania was excluded from this research, mainly because in population size as well as income status it is dominated by Australia and New Zealand. Because of this heterogeneity it was impossible to select representative countries and territories. Australia and New Zealand, as well as Singapore and the Chinese territory Hong Kong, started with a zero-COVID policy, resulting in relatively few infections and deaths as compared to the rest of the world [18,19]. During or after the omicron outbreak these countries shifted to a living-with-COVID policy, which resulted in a (temporary) increase in percent excess mortality [20].

At first glance it is counterintuitive to find a positive correlation between the GDP of a given country or territory and the number of SARS-CoV-2 infections (Figure 2) [21]. It would be expected that the higher the GDP, the more advanced the health care system, higher vaccination rates, and the possibility to impose strict societal restrictions. On the other hand, the number of recorded infections also depends on the intensity and degree of testing [22,23]. Low-income countries will not have the medical infrastructure and/or the financial means to be able to afford elaborate testing for the virus. It was reported by medical experts that in Africa, the number of reported cases is an acute underestimation, which can be attributed to the poor African health systems, a lack of sufficient test kits, and inadequate laboratory capacity [24]. Several studies were performed on the number of COVID-19 cases that remain undetected. In October 2021, the WHO calculated that the number of actual cases in Africa was seven times higher than the detected numbers [25]. A similar study found that the actual number of cases in the European countries Italy, Portugal, and Switzerland was four times higher in April 2020 [26]. Furthermore, the WHO estimated that there were 2.74 times more SARS-CoV-2 deaths globally in 2020 and 2021 than reported [27]. It is expected that the number of undetected cases in European countries is lower in October 2021 than in April 2020, because countries had more time to anticipate in this time period. However, to draw valid conclusions about the number of undetected cases, data from the same period needs to be obtained. Furthermore, it is also still dependent on the willingness of people with symptoms to get tested or not [28].

The effect of the difference in Government Stringency Index (GSI) (maximal GSI-GSI April 2022) on the three virus variants shows that the higher this difference, the larger the number of omicron infections (Figure 4). For most countries, the maximal GSI was imposed during the first wave and during the delta wave, after which governments decided to loosen the restrictions in the summer of 2021, when infection rates decreased in most countries. This led to an explosion of omicron infections in late 2021 and early 2022 [29,30]. As mentioned above, the highest GSI rates were reached during either the first wave or the delta wave. A rebound effect, as seen with omicron, was not observed when the delta variant emerged.

Implementation and feasibility of strict societal restrictions differ for high-income and low-income countries. For instance, in a country like India, strict government policies may not lead to better compliance but can overwhelm the healthcare systems [31].

Relief of societal restrictions was associated with a surge in omicron infections, and the number of infections was highest in countries with the least restrictions. The high replication number of omicron could explain this effect [32,33]. For some countries, the strictness of the GSI was dependent on vaccination status, meaning that vaccinated people were allowed more social contacts, including traveling [34,35]. Using a cut-off of a 20-percent points difference in GSI, the countries with (temporary) relief of societal restrictions for vaccinees included Pakistan, South Africa, France, Denmark, Italy, and the United States. The available data do not allow us to conclude whether such a policy influenced the control of spread of SARS-CoV-2 in the population. Implementation of societal restrictions has been used as a measure to control COVID-19. It is possible that limiting the spread of the virus would also reduce the emergence of new variants.

Regarding vaccination rate, regardless of the type of vaccine used, the vaccines offered better protection against the delta variant than against the omicron variant. This finding is reported in previous studies as well [36,37]. For each continent, a negative correlation was found between vaccination rate and rate of delta infections. For omicron infections this was not always the case. Furthermore, if a negative correlation was found between vaccination rate and omicron infections, this correlation was always found to be weaker than for the delta infections. However, as mentioned above, this effect could also be attributed to the intensity of the GSI during that wave. For most countries, the GSI was higher at the time of the delta wave than at the time of the omicron wave.

It is known that the omicron variant has a higher R_0_ than delta, and therefore the number of infections can be expected to be higher, irrespective of vaccination status or GSI [32,38,39]. On the other hand, omicron has a lower morbidity and mortality rate than delta, a finding confirmed on all continents [40,41,42,43,44]. Our analysis is restricted to the number of infections, and not hospitalization or COVID-19 related deaths. Analyzing hospitalization and COVID-19 related deaths would give a subjective view on the matter, because this is dependent on the quality of the healthcare system in a country.

The analyses show that the strongest trends, and thus the highest R^2^ values, were found in analyses related to the omicron variant. The most plausible explanation for this is the omicron wave being the last wave, with the most reliable data. Countries had more time to set up testing facilities and other infrastructure during the omicron variant than in an earlier phase of the pandemic. This implies that infection data is most reliable for the omicron variant.

In high-income countries, the potential effects of a COVID pass on spread of SARS-CoV-2 could not be determined because all countries included in the analysis implemented such a pass. Analysis and comparison of low- and middle-income countries did not show any effect of the COVID pass on infections (Figure 5). The COVID pass had different prerequisites and privileges across countries, and there is no clear overview for which activities a COVID pass was required, and over what time period it was required in each country. A COVID pass could still be required for travelers upon entering a country, territory, or region where SARS-CoV-2 is strictly controlled with close to zero infections.

There are several limitations to the present study. First of all, all of the data was collected from the same source. This means that potential structural bias or mistakes in data collection would be present in all the data used in the study. Furthermore, for some countries it can be argued how reliable the published data is, for instance regarding case definition. All analyses were performed country-wise and not patient-wise. This means that the study did not use data from individual patients, and underlying conditions or co-morbidities were not accounted for. For assessment of the vaccination rate, all types of vaccines were included, and no sub-analysis was performed on the differential efficacy of certain vaccine types. Furthermore, the present study considered what was at first established as fully vaccinated, which is at two doses, whereas now, fully vaccinated could also be considered two doses and an additional booster dose. The number of undetected cases is also a limitation to this study, especially in countries with high immigration rates where documented vaccination rates and infection rates might be deviant from the actual rates. For the COVID pass, it was very hard to find reliable sources, and within the different passes, there are different guidelines. Whereas some countries only used the COVID pass at the border to accept or deny people from entering the country, some countries also used it to accept or deny people from entering buildings or events. Furthermore, requirements for a valid COVID also differed per country.

A limitation of the analysis of the burden of COVID-19 at the country level is that minority groups (either because of socio-economic position, age, access to health care, or otherwise) could be disproportionally affected [45,46].

At a press conference in March 2023, the Director-General of the WHO, Dr Tedros Adhanom Ghebreyesus, indicated that he was confident that in 2023 COVID-19 will be over as a public health emergency of international concern [47] and on May 23, 2023, the official end of the pandemic was declared [48]. SARS-CoV-2 will remain endemic, infecting susceptible individuals in the populations worldwide. It can be expected that the current variant(s) of SARS-CoV-2 have or will obtain the characteristics of other beta coronaviruses such as HKU1 and OC43, causing mild upper respiratory tract infections during the winter season.

## 5. Conclusions

Our data show that across the world, regardless of the vaccine used, the vaccines offered better protection against the delta variant than against the omicron variant of SARS-CoV-2. A positive correlation was found between the gross domestic product of a country or territory and the number SARS-CoV-2 infections. This is likely due to the fact that high-income countries have better medical infrastructure and the financial means to test and thereby document the spread of the virus. Furthermore, the lifting of societal restrictions (expressed as reduction of Government Stringency Index) was associated with the number of omicron infections. Relief of societal restrictions was accompanied by a rise in omicron infections, the highest number of infections found in countries and territories with the least restrictions. Finally, analysis and comparison of low and middle-income countries did not show any effect of the use of a COVID pass on SARS-CoV-2 infections.

## Figures and Tables

**Figure 1 vaccines-11-01407-f001:**
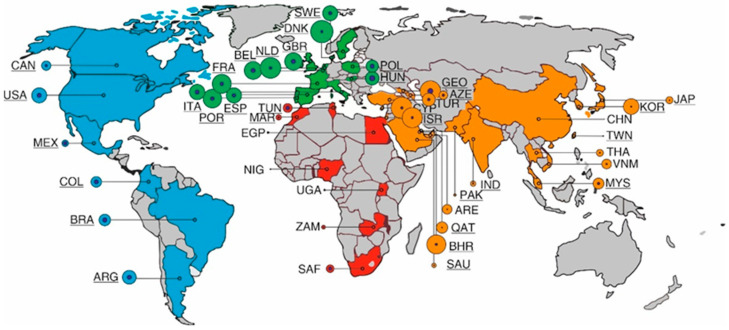
Overview of countries and territories included in this study, color-coded by continent and indicated by their 3-letter abbreviation (https://www.iban.com/country-codes (accessed on 31 March 2022)).Countries which have used a COVID pass are underlined. The number of COVID-19 cases (per 100,000) is indicated by proportional circles; inner circles represent the relative number of COVID-19 deaths (per 1000,000).

**Figure 2 vaccines-11-01407-f002:**
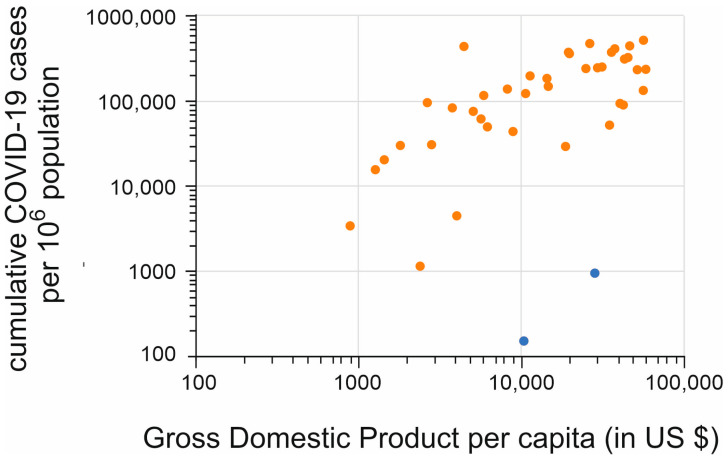
Correlation between Gross Domestic Product and the (cumulative) number of COVID-19 cases. R^2^ = 0.3015, F = 16.84, *p* < 0.01. Orange symbols represent individual countries or territories. China and Taiwan are indicated with blue symbols.

**Figure 3 vaccines-11-01407-f003:**
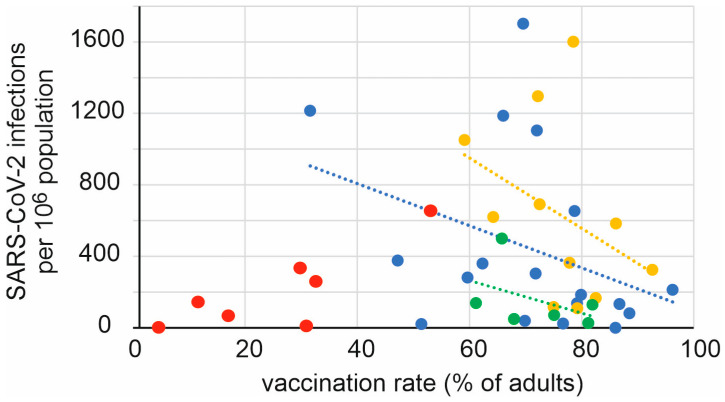
Correlation between vaccination rate and number of SARS-CoV-2 infections per 10^6^ population. Individual countries or territories are color-coded by continent: Asia blue, Africa red, Europe yellow, and the Americas green. Linear regression lines are indicated as dotted lines in the color of the respective continent. The correlation coefficients R^2^ for linear regression were 0.137 (Asia), 0.138 (Europe), and 0.195 (America); *p* < 0.01 in all cases. With the exception of Tunisia, all other African countries had vaccination rates below 40%.

**Figure 4 vaccines-11-01407-f004:**
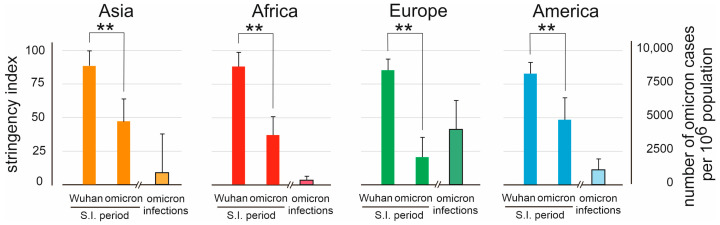
Stringency index (S.I.) of societal restrictions during the period that the SARS-CoV-2 Wuhan strain was dominant (March 2020 to September 2020) as compared to the period when omicron was dominant (analyzed from December 2021 to March 2022) (plotted on the left *Y*-axis). Mean + S.D. of the countries or territories included per continent is shown. The number of omicron infections per 10^6^ population from December 2021 to March 2022 is shown (plotted on the right *Y*-axis). ** indicates significant difference (*p* < 0.01) in two-sided Student’s *t*-test.

**Figure 5 vaccines-11-01407-f005:**
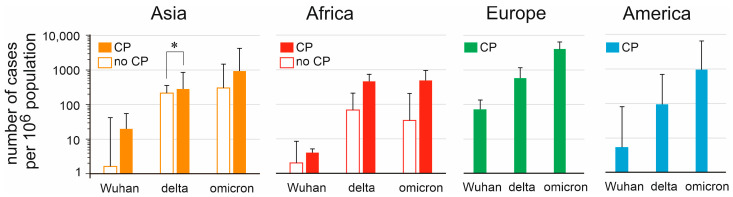
Relation between implementation of a COVID pass (CP) and number of COVID-19 cases (expressed per 10^6^ population) during the first wave of SARS-CoV-2 (Wuhan strain), the delta wave, and the omicron wave. Of the Asian countries or territories included in this study, 12/17 did use a CP, in Africa this was 2/7, while in Europa and the Americas each country included in the study used a CP of some sort. * indicates significant difference (*p* < 0.05) in two-sided Student’s *t*-test.

**Table 1 vaccines-11-01407-t001:** Demographics and burden of COVID-19. Countries or territories are listed in alphabetical order per continent. Population size per 2020 was obtained from https://www.worldometers.info/world-population/population-by-country/ (accessed on 31 March 2022); for India from https://www.macrotrends.net/countries/IND/india/population (accessed on 31 March 2022). GDP is Gross Domestic Product per capita, expressed in US$; HDI is Human Development Index.

Continent and Country or Territory	Population (as of 2020) [16,17]	Cases (Cumulative per Million People; 31 March 2022)	Deaths (Cumulative per Million People; 31 March 2022)	Vaccination Rate (per 31 March 2022) in Percentage	GDP per Capita (US$) 2020	HDI
Asia						
Azerbaijan	10,139,177	76,450	935	47.19	5083	756
Bahrain	1,701,575	376,128	998	69.56	19,514	852
China	1,439,323,776	155	3	86.02	10,370	761
Cyprus	1,207,359	475,863	1046	72.02	26,372	887
Georgia	3,989,167	440,071	4470	31.59	4447	812
India	1,396,378,127	30,360	367	59.66	1811	645
Israel	8,655,535	413,196	1109	65.99	37,488	919
Japan	126,476,461	52,547	226	79.86	34,813	919
Malaysia	32,365,999	123,263	1029	78.74	10,631	810
Pakistan	220,892,340	20,618	129	51.41	1446	557
Qatar	2,881,053	134,050	251	88.5	56,026	848
Saudi Arabia	34,813,871	29,563	248	69.88	18,691	854
South Korea	51,269,185	252,734	313	86.73	31,327	916
Taiwan	23,816,775	969	36	76.66	28,306	949
Thailand	69,799,978	50,222	349	71.8	6198	777
Turkey	84,339,067	173,963	1148	62.3	12,038	820
United Arab Emirates	9,890,402	94,372	243	96.22	40,438	890
Vietnam	97,338,579	96,471	423	79.15	2655	704
Africa						
Egypt	102,334,404	4,552	220	30.89	4028	707
Morocco	36,910,560	31,053	428	32.56	2818	686
Nigeria	206,139,589	1,168	14	4.52	2396	539
South Africa	59,308,690	62,060	1669	29.81	5659	709
Tunisia	11,818,619	83,835	2292	53.2	3780	740
Uganda	45,741,007	3468	76	17.01	891	544
Zambia	18,383,955	15,813	198	11.62	1273	584
Europe						
Belgium	11,589,623	326,663	2638	78.52	45,189	931
Denmark	5,792,202	519,331	963	82.5	56,202	940
France	65,273,511	376,011	2097	77.84	35,785	901
Hungary	9,660,351	185,528	4558	59.13	14,368	854
Italy	60,461,826	246,758	2697	79.23	29,359	892
Netherlands	17,134,872	446,569	1255	72.17	46,345	944
Poland	37,846,611	149,482	2886	64.23	14,660	880
Portugal	10,196,709	362,778	2097	92.6	19,771	864
Spain	46,754,778	241,981	2149	86.09	24,939	904
Sweden	10,099,265	235,250	1724	74.98	51,539	945
United Kingdom	67,886,011	313,253	2846	72.49	43,020	932
America						
United States	331,002,651	237,189	2905	65.77	58,203	926
Canada	37,742,154	90,526	977	81.92	42,258	929
Colombia	50,882,891	117,294	2691	67.93	5892	767
Brazil	212,559,417	138,962	3064	75.1	8228	765
Argentina	45,195,774	198,529	2812	81.18	11,334	845
Mexico	128,932,753	44,346	2532	61.19	8909	779

## Data Availability

Not applicable.

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
