# Peer review of "The Impact of SARS-CoV-2 Immune Status and Societal Restrictions in Controlling COVID-19 across the World"

_vaccines, 2023, doi:10.3390/vaccines11091407_

Round 1

Reviewer 1 Report

Comments for the authors

1.        In the abstract part, it would be better to elucidate the significance of the presentation of "A global analysis of SARS-CoV-2 immune status (determined by past infection and/or vaccination), vaccination rates, as well as societal restrictions in controlling the COVID-19 pandemic.". 

2.        Beside the possible controlling the spread of COVID-19, could it prohibit the possible the emergence of new variants?

3.        Does the application of COVID pass provide any clues for emerging infectious diseases?

4.        Chinese Taiwan or Taiwan China; Chinese Hong Kong or Hong Kong China should be highly suggested to take place of "Taiwan" or "Hong Kong".

5.        Line 12, the authors mentioned "or a recent (24-48 hours) negative SARS-CoV-2 antigen test", does it mean that it is only negative for antigen, without requiring vaccination or recovering from past infection?

6.        Line 13-14, "A global analysis of SARS-CoV-2 immune status 13 (determined by past infection and/or vaccination),", it would be more formal to delete the bracket here.

7.        Line 57, it would be more formal to change "180+ countries" into "more than 180 countries.

8.        In the Methods part, it would be better to present the total number of these countries and territories.

9.        Line 74, For consistency, "in Europe" should be "Europe".

10.     In the table 1, the unit of each column should be added in the first row. "Vaccination rate (per March 31, 2022)", is the unit of it % ?

11.     Figure 2 legend, what does the deep yellow color represent?

12.     Figure 3, Africa red, I am sure it is red or dark yellow.

13.     line 181, is it "omega" not omicron?

14.     In the Figure 4, it is omicron infections on the right X-axis and, but in figure 4 legend, it shows as omega infections. and in figure 4 legend, it is X-axis, but not Y-axis.

15.     Figure 1, how or where to get the information of COVID pass in these countries or territories?

16.     Figure 5, for consistency, the column charts should be arranged as Afria, Asia, Europe and America from left to right.

17.     Line 218-220, these opinions are reasonable, but according to the common sense in come country or territory,

18.     COVID pass prefers to be required, when in the situation like this: the infection of SARS-CoV-2 infections are strictly controlled (almost zero infection) in some district, the foreign travelers want to visit this zero infection district.

19.     Line 222, the SARS-Cov-2 should be "SARS-CoV-2".

20.     Line 238, the bracket of (also) should deleted.

21.     Line 295, according to the analysis that the infection data is most reliable for the omicron,

22.     The authors could highlight this related data and make some conclusions based on these data.

23.     In the discussion part, when the authors discussed data from figures or tables, which should be shown in the brackets.

24.     It would better to introduce the most recent references.

Author Response

At first, we would like to thank both reviewers for their detailed, critical, and constructive remarks. A detailed response to each remark is given below.

Their comments have really helped us to improve our manuscript, which we hope would now be acceptable for publication. We are looking forward to your evaluation.

Reviewer 1 Comments for the authors

  1. In the abstract part, it would be better to elucidate the significance of the presentation of "A global analysis of SARS-CoV-2 immune status (determined by past infection and/or vaccination), vaccination rates, as well as societal restrictions in controlling the COVID-19 pandemic.".

Reviewer correctly concludes that we have not included our significant findings in the Abstract. In the revised version we have added: “The data show that across the world, vaccination was more effective in reducing SARS-CoV-2 infections with the delta variant than the omicron variant. Strict societal restrictions could control spread of the virus, but relief of the restrictions was associated with an increase in omicron infections. No significant in SARS-CoV-2 infections were found when comparing countries or territories which did or did not implement a COVID pass.” (lines 15-20 revised manuscript)

  1. Beside the possible controlling the spread of COVID-19, could it prohibit the possible the emergence of new variants?

This is an excellent remark! Yes, controlling the spread would also reduce the emergence of new variants. We have added a sentence to the relevant part in the Discussion (lines 283-285 revised Discussion).

  1. Does the application of COVID pass provide any clues for emerging infectious diseases?

This is difficult to establish with the available data. All European and American countries included in our study did use a COVID pass for a certain period. The societal restrictions certainly had an effect on other respiratory diseases such as pertussis. An extensive discussion of this issue is tempting, but would be outside the main scope of our paper.

  1. Chinese Taiwan or Taiwan China; Chinese Hong Kong or Hong Kong China should be highly suggested to take place of "Taiwan" or "Hong Kong".

We have referred to Taiwan and Hong Kong as the Chinese territory Taiwan (line 78) and Chinese territory Hong Kong (line 237) where appropriate.

  1. Line 12, the authors mentioned "or a recent (24-48 hours) negative SARS-CoV-2 antigen test", does it mean that it is only negative for antigen, without requiring vaccination or recovering from past infection?

Yes, we mean a negative SARS-CoV-2 antigen test. We have made it clear by editing the complete sentence as follows: A COVID pass could be obtained when either fully vaccinated against COVID-19, or having re-covered from a documented COVID-19 episode, or a recent (24-48 hours) negative SARS-CoV-2 antigen test. (lines 11-12).

  1. Line 13-14, "A global analysis of SARS-CoV-2 immune status 13 (determined by past infection and/or vaccination),", it would be more formal to delete the bracket here.

The brackets have been deleted as suggested (line 14).

  1. Line 57, it would be more formal to change "180+ countries" into "more than 180 countries.

We have changed this sentence into  . . .  more than 180 countries or territories (line 62).

  1. In the Methods part, it would be better to present the total number of these countries and territories.

We have added in the Methods that we analyzed 42 countries and territories (line 76)

  1. Line 74, For consistency, "in Europe" should be "Europe".

Agree! Has been changed into “Euope” (line 82).

  1. In the table 1, the unit of each column should be added in the first row. “Vaccination rate (per March 31, 2022)”, is the unit of it % ?

Yes, the unit is % and this has been added to the Table.

  1. Figure 2 legend, what does the deep yellow color represent?

Orange symbols represent individual countries or territories, and the legend has been revised accordingly.

  1. Figure 3, Africa red, I am sure it is red or dark yellow.

We agree, the red color of the symbols for African countries were rather dark yellow. We have replaced the symbols for clear red ones.  

  1. line 181, is it “omega” not omicron?

Of course it is omicron; mistake!

  1. In the Figure 4, it is omicron infections on the right X-axis and, but in figure 4 legend, it shows as omega infections. And in figure 4 legend, it is X-axis, but not Y-axis.

Same mistake as above; have corrected that in the legend. The number of infections are plotted on the right Y-axis, while the Stringency Index is plotted on the left Y-axis.

  1. Figure 1, how or where to get the information of COVID pass in these countries or territories?

The data on use of a COVID pass where obtained from the official national health institutes from the respective countries or territories. When such information was not provided, secondary sources for such country or territory were used. This information is given in the Methods section, lines 103-106.

  1. Figure 5, for consistency, the column charts should be arranged as Africa, Asia, Europe and America from left to right.

We agree that there was inconsistency in the order of the continents. We now have revised all figures, so that the order as in Table 1 (Asia, Africa, Europe, and America) is maintained throughout the Figures and the manuscript. Also in the Methods, this order of continents is used (lines 76-84).

  1. Line 218-220, these opinions are reasonable, but according to the common sense in some country or territory,

We agree, these are general remarks and opinions.

  1. COVID pass prefers to be required, when in the situation like this: the infection of SARS-CoV-2 infections are strictly controlled (almost zero infection) in some district, the foreign travelers want to visit this zero infection district.

We agree and have added a sentence to the Discussion: A COVID pass could still be required for travelers upon entering a country, territory, or region where SARS-CoV-2 is strict controlled with close to zero infections. (lines 314-315)

  1. Line 222, the SARS-Cov-2 should be "SARS-CoV-2".

               This has been corrected

  1. Line 238, the bracket of (also) should deleted.

               We have removed the brackets as suggested.

  1. Line 295, according to the analysis that the infection data is most reliable for the omicron,

  1. The authors could highlight this related data and make some conclusions based on these data.

We believe we have discussed this aspect in lines 302-307 of the Discussion.

  1. In the discussion part, when the authors discussed data from figures or tables, which should be shown in the brackets.

This is an excellent suggestion and we have included the relevant Figure numbers at appropriate places in the Discussion.

  1. It would better to introduce the most recent references.

We have done our best to include the most relevant as well as the most recent references in our manuscript and 12 out of the 48 references are from 2023.

Reviewer 2 Report

Thank you for sharing your article on control measures of COVID-19. Here some comments that could help to improve the article:

L61-64: Please clearly state whether you used the OxGRT database for your research. Please rephrase in your manuscript as appropriate. 

L67-68: What is your rationale for including the countries listed in L71-76? Please incorporate in your manuscript. L118-119: Is the availability of reliable data your rationale for including those countries? Please revise your manuscript accordingly. 

L121: Cases and deaths based on which data source(s)? 

Table 1: Please state in your manuscript how cases and deaths were confirmed. Also, are those deaths only due to COVID-19 or COVID-19 & co-morbidities? Regarding vaccination, is it know which vaccine(s) were administered and how many times? 

L65: Please clearly state throughout your methods section which database(s) you used for your research and when you accessed it. 

See above.

Author Response

At first, we would like to thank both reviewers for their detailed, critical, and constructive remarks. A detailed response to each remark is given below.

Their comments have really helped us to improve our manuscript, which we hope would now be acceptable for publication. We are looking forward to your evaluation.

Reviewer 2

Thank you for sharing your article on control measures of COVID-19. Here some comments that could help to improve the article:

L61-64: Please clearly state whether you used the OxGRT database for your research. Please rephrase in your manuscript as appropriate.

We have restated in the Methods that the data were derived from the OxGRT database: The data was collected from the OxGRT database and analyzed per continent,  .. . .  (line 73)

L67-68: What is your rationale for including the countries listed in L71-76? Please incorporate in your manuscript. L118-119: Is the availability of reliable data your rationale for including those countries? Please revise your manuscript accordingly.

We have revised the relevant sentence in the Methods as follows: “The data was collected and analyzed per continent, of which the selection of individual countries and territories was determined based on the coverage of the HDI and availability of sufficient and reliable data”. (lines 73-75 revised manuscript). Now the Methods section is in line with the first part of the Results.

L121: Cases and deaths based on which data source(s)?

Table 1: Please state in your manuscript how cases and deaths were confirmed. Also, are those deaths only due to COVID-19 or COVID-19 & co-morbidities? Regarding vaccination, is it know which vaccine(s) were administered and how many times?

The data on cases and deaths was collected from the OxGRT database. The case definition of the WHO may or may not have been used by every country but there is no way to confirm that. The same holds true for co-morbidities. Many, but not all, COVID-19 patients have co-morbidities. The available data did not allow to provide further details. We have included above factors as limitations of our study: “Furthermore, for some countries it can be argued how reliable the published data is, for instance regarding case definition. All analyses were performed country-wise and not patient-wise. This means that the study did not use data from individual patients, and underlying conditions or co-morbidities were not accounted for”. (line 318-322)

The type of vaccines used will vary between countries, but also within a given country or territory, different types of vaccines have been used. Also depending on the type of vaccine used, a single dose or multiple doses will have been used.

L65: Please clearly state throughout your methods section which database(s) you used for your research and when you accessed it.

We have re-checked the Methods and included the databases used and the time and period covered in the analysis.

Round 2

Reviewer 1 Report

All the issues I  put forward are well addressed by the authors. 

Reviewer 2 Report

Thank you for sharing the revised manuscript. My comments and suggested edits were addressed sufficiently. 

Please see above.